# Multisystem Inflammatory Syndrome in Children Following COVID-19 Vaccination: A Sex-Stratified Analysis of the VAERS Database Using Brighton Collaboration Criteria

**DOI:** 10.3390/ph16091231

**Published:** 2023-08-30

**Authors:** Valerio Liguori, Alessia Zinzi, Mario Gaio, Consiglia Riccardi, Luigi Di Costanzo, Francesca Gargano, Claudia Carpentieri, Maria D’Elia, Francesca Futura Bernardi, Ugo Trama, Annalisa Capuano, Concetta Rafaniello

**Affiliations:** 1Campania Regional Centre for Pharmacovigilance and Pharmacoepidemiology, 80138 Naples, Italy; valerio.liguori@unicampania.it (V.L.); mario.gaio@unicampania.it (M.G.); consiglia.riccardi@unicampania.it (C.R.); annalisa.capuano@unicampania.it (A.C.); concetta.rafaniello@unicampania.it (C.R.); 2Section of Pharmacology “L. Donatelli”, Department of Experimental Medicine, University of Campania “Luigi Vanvitelli”, 80138 Naples, Italy; luisdico86@gmail.com; 3Regional Direction for Health Management, Pharmaceutical Unit, 80131 Naples, Italy; f.gargano@policlinicocampus.it (F.G.); claudiacarpentieri1995@gmail.com (C.C.); mdelia@unisa.it (M.D.); bernardi.francesca.futura@gmail.com (F.F.B.); ugo.trama@regione.campania.it (U.T.)

**Keywords:** MIS-c, children, COVID-19 vaccines, SARS-CoV-2, VAERS, safety monitoring, sex differences

## Abstract

Multisystem inflammatory syndrome in children (MIS-c) is an uncommon, but serious, inflammatory response that occurs after SARS-CoV-2 infection. As time went by, MIS-c was also reported as a potential adverse event following COVID-19 vaccination. A descriptive analysis was performed of Individual Case Safety Reports (ICSRs) associated with anti COVID-19 vaccines and related to the pediatric population from 2020 to 2022. The present pharmacovigilance study aimed to describe cases of MIS-c following COVID-19 vaccination, stratified by sex, reported in the Vaccine Adverse Events Reporting System (VAERS) and meeting the Brighton Collaboration criteria for case definition. We assessed all suspected cases through the case definition and classification of the Brighton Collaboration Group, and only definitive, probable, and possible cases were included in the analysis. The Reporting Odds Ratio (ROR) with 95% Confidence Interval (CI) was computed to assess if males have a lower/higher probability of reporting ICSRs with MIS-c compared with females. Overall, we found 79 cases of potentially reported MIS-c following vaccination. This study demonstrated that MIS-c following vaccination was more commonly reported for male subjects with a median age of 10 years (IQR 10.0–11.4), especially after the first dose of anti COVID-19 vaccines with a median time to onset of 27 days. Even so, the rate of occurrence of MIS-c following anti COVID-19 vaccines is lower (0.12/100,000 vaccinated subjects; 95% CI, 0.12–0.13). Overall, all ICSRs were serious and caused or prolonged hospitalization. Finally, disproportionality analysis showed that males had a higher reporting probability of MIS-c compared with females following immunization with mRNA COVID-19 vaccines. Since only a few years of marketing are available, further data from real-life contexts are needed.

## 1. Introduction

During the COronaVIrus Disease 2019 (COVID-19) pandemic, an increased number of cases of Multisystem Inflammatory System in children (MIS-c) has been observed following anti COVID-19 vaccination by the Centers for Disease Control and Prevention (CDC) [1]. At the same time, in September 2021, the European Medicines Agency’s (EMA) Pharmacovigilance Risk Assessment Committee (PRAC) started the review of suspected MIS reported after the administration of anti COVID-19 vaccines in the European Economic Area (EEA) [2]. While MIS-c following Severe Acute Respiratory Syndrome–Coronavirus 2 (SARS-CoV-2) infection is well-known, little is known about COVID-19 vaccine-induced MIS-c. Only a few cases of MIS-c have been reported in the literature in subjects who had no history of SARS-CoV-2 infection [3,4,5,6,7]. These cases suggest that a possible association between the COVID-19 vaccine and MIS-c cannot be excluded, but is not yet confirmed; moreover, the pathophysiology of MIS-c is unknown and it is rarely reported.

MIS-c represents a late manifestation of SARS-CoV-2 infection which can occur 2–6 weeks after SARS-CoV-2 infection or exposure [8]. MIS-c following COVID-19 vaccination is challenging to diagnose due to a lack of diagnostic biomarkers [7]. Moreover, it is expected that some cases of MIS-C caused by SARS-CoV-2 infections acquired prior to full vaccination will occur after vaccination and will appear to be temporally associated with the vaccine. Also, some MIS-c cases caused by SARS-CoV-2 infection might occur if protection against infection is incomplete [1].

Some of the clinical manifestations of MIS-c are similar to other known medical conditions, mainly Kawasaki Disease (KD) and Toxic-Shock Syndrome (TSS) and macrophage activation syndrome (MAS)/secondary hemophagocytic lymphohistiocytosis (HLH) [9]. However, although the pathophysiology of this illness is unknown, previous studies showed that MIS-c is characterized by a cytokine storm associated with a superantigen-like activation of T cells with an expansion of Vβ21.3-expressing T cells which is not seen in KD, TSS and MAS/HLH [10,11,12,13].

Persistent fever is the predominant symptom in MIS-c (usually persisting for 24 h and presenting for several days prior to diagnosis). Furthermore, gastrointestinal symptoms (e.g., stomach pain, diarrhea and vomiting), dermatological/mucocutaneous symptoms (e.g., skin rash or conjunctivitis), cardiovascular symptoms (e.g., myocardial dysfunction or arrhythmia), respiratory symptoms (e.g., shortness of breath or pneumonia), neurologic symptoms (e.g., headache) and renal symptoms (e.g., acute insufficiency) are also frequent within MIS-c [14]. Currently, diagnosis of MIS-c is based on laboratory tests (e.g., blood and urine tests) and imaging tests, such as a chest X-ray (CXRk), echocardiogram, abdominal ultrasound or computed tomography (CT) [15]. Several studies estimate the MIS-c incidence is approximately 2 per 100,000 persons younger than 21 years old [16,17]. A Danish nationwide prospective cohort study on children and adolescents (aged 0–17 years) estimated that the incidence of MIS-c was 1 in 3400 unvaccinated individuals (95% CI 2600–4600) and 1 in 9900 vaccinated individuals (95% CI 1800–390,000) with breakthrough infection [18]. Furthermore, a case series on vaccinated individuals aged 12–20 years estimated that the overall reporting rate for MIS-c following anti COVID-19 vaccination was 1 per 1,000,000 persons receiving ≥1 vaccine dose [1]. MIS-c was first described in the United Kingdom in April 2020, a few weeks after the infection of SARS-CoV-2 [19]. Afterwards, more than 9000 cases of MIS-c have been reported in the United States (US) by the CDC as of 30 January 2023 [20]. No specific risk group has been found; however, the median age of children diagnosed with MIS-c is 8–11 years and the majority of patients are predominantly male [8,21]. In this regard, considering that MIS-c has only recently been identified as a rare adverse event following anti COVID-19 vaccination, monitoring post-vaccine MIS on special populations it’s important to better define the benefit-risk profile of anti COVID-19 vaccines. Therefore, we provide a post-marketing evaluation of the safety of anti COVID-19 vaccines focusing on the potential cases of MIS in the paediatric population, through the analysis of the VAERS database. We also aimed to assess the Reporting Odds Ratio (ROR) and 95% Confidence Interval (CI) to evaluate if males have a lower/higher probability of reporting Individual Case Safety Reports (ICSRs) with MIS-c in a direct comparison with females. Given the data source, the contribution of vaccination to MIS-c in individuals is unknown and cannot be estimated based on our surveillance data.

## 2. Results

Throughout the study period, a total of 1,038,616 ICSRs were obtained from VAERS, 164 of which reported MIS-c as an adverse event following anti COVID-19 vaccinations and were associated with subjects aged < 21 years. Through the application of Brighton Collaboration’s algorithm, 85 ICSRs were excluded (N = 58 ICSRs with insufficient evidence; N = 27 ICSRs with no MIS-c reported). Of the 79 remaining cases of potentially reported MIS-c, N = 49 resulted as “Definitive” cases (Level 1), N = 26 as “Probable” cases (Level 2) and N = 4 as “Possible” cases (Level 3) (Figure 1).

Looking at the included ICSRs stratifying by sex, our analysis showed that MIS-c was more commonly reported for male subjects (N = 45; 57%). As reported in Table 1, the median age of patients who experienced MIS-c was 11 years (IQR 8–11.1). Hospitalization was the most frequently reported outcome (N = 60; 75.9%); the length of hospitalization was 4 days (IQR 2–6), as the median. BNT162b2 mRNA COVID-19 vaccine was nearly the only vaccine reported as suspected (N = 77; 97.4%) and MIS-c mainly occurred after the first dose of vaccination (N = 55; 69.6%); regardless of vaccine dose, the median time to onset was 27 days (IQR = 5.0–59.1) (Table 1).

As shown in Figure 2, most cases of MIS-c following anti COVID-19 immunization were mainly observed in subjects aged between 6 and 15 years (N = 63; 79.7%). More in detail, stratified analysis by sex, our analysis showed that MIS-c was more commonly reported for male subjects (N = 28; 35.5%) aged 11–15 years (Figure 2).

As regards the AEFIs distribution according to MedDRA’s SOCs, the analysis of included ICSRs showed that the most common SOC was “Investigations” (N = 989; 48.8%), followed by “Gastrointestinal disorders” (N = 145; 7.2%), “General disorders and administration site conditions” (N = 142; 7.0%), “Cardiac disorders” (N = 102; 5.0%), “Surgical and medical procedures” (N = 91; 4.5%), “Infections and infestations” (N = 83; 4.1%), “Respiratory, thoracic and mediastinal disorders” (N = 80; 3.9%), and “Immune system disorders” (N = 71; 3.5%) while the other SOCs accounted for less than 3%. More in detail, looking at the type of events, within each SOC category, a specific AEFI was more commonly reported than others. In this regard, “C-reactive protein increased” was the most frequently reported AEFI within Investigations, “nausea and vomiting symptoms” within Gastrointestinal disorders, “pyrexia” within General disorders and administration site conditions, “left ventricular dysfunction” within Cardiac disorders, “immunoglobulin therapy” within Surgical and medical procedures, “COVID-19” within Infections and infestations, “cough” within Respiratory, thoracic and mediastinal disorders, as well as “Multisystem inflammatory syndrome in children” within Immune system disorders (Table 2). As reported in Table 2, AEFI’s distribution did not show sex differences between male and female subjects.

Excluding SOC Investigations, we analyzed the most frequently reported PTs (excluding PT Multisystem inflammatory syndrome in children). As expected, pyrexia was the most commonly reported AEFI (N = 60, 9.4%), followed by immunoglobulin therapy (N = 45, 7.0%), abdominal pain (N = 37, 5.8%), vomiting (N = 35, 5.5%), headache (N = 32, 5.0) and diarrhoea (N = 31, 4.8%) (Figure 3).

Furthermore, we analyzed the most frequently reported PTs stratifying the data by sex (Figure 4). Our analysis showed greater cardiac involvement for males than females. Specifically, PTs more commonly reported chest pain, left ventricular dysfunction, tachycardia, coronary artery dilatation and sinus tachycardia.

Looking at the probability of reporting ICSRs with MIS-c, males were associated with a higher risk of reporting MIS-c (ROR = 3.07, 95% CI 2.24–4.20, *p* << 0.005) compared to females. These results are confirmed in the male group after the application of the algorithm of Brighton Collaboration (ROR = 2.69, 95% CI 1.72–4.20, *p* << 0.005) (Table 3).

Finally, it was also possible to calculate the rate of incidence of reporting anti COVID-19-related MIS-c by retrieving data about the total US population which received at least one dose of a COVID-19 vaccine from Our World In Data (https://ourworldindata.org/covid-vaccinations, accessed on 31 January 2023). In this regard, it was estimated the rate of reporting MIS-c associated with COVID-19 vaccination was equal to 0.12 per 100,000 vaccinated subjects (95% CI, 0.12–0.13).

## 3. Discussion

In this study, from 1 January 2020 to 31 December 2022, we analyzed all ICSRs with MIS-c reported following anti COVID-19 vaccination through the Vaccine Adverse Event Reporting System (VAERS), the US national vaccine passive surveillance system. In order to provide a descriptive analysis of this event with a focus on possible sex-related differences, we included all cases that met the case definition and classification of the Brighton Collaboration for MIS-c [19]. During the study period, 1,038,616 ICSRs were analyzed and 79 met the case definition of the Brighton Collaboration for MIS-c. Even if they met the case definition of the Brighton Collaboration, we are not able to establish a causal relationship between the vaccine and MIS-c. Vaccination’s contribution to MIS-c in individuals is unknown and cannot be assessed using surveillance data. It is also possible that some of these 79 individuals had other unrecognized inflammatory conditions, or might have been infected with SARS-CoV-2 in the recent past, and vaccination might be coincidental to the subsequent MIS-c. Moreover, nearly 40% of included ICSRs (30 of 79 cases) are classified as “Possible” or “Probable”, implying that a definitive diagnosis has not been provided.

As expected, most of these ICSRs (97.4%) were associated with the BNT162b2 mRNA COVID-19 vaccine. Furthermore, regarding the demographic characteristics, most of the identified cases occurred in males younger than 21 years of age after the first dose of vaccination. Our results are consistent with those of previous real-word studies, which highlighted that MIS-c was more frequently observed in younger males after the primary vaccination course with anti COVID-19 vaccines [1,22,23,24]. Specifically, in line with Yousaf AR et al., who also used the Brighton Collaboration’s case definition and classification for MIS-c, this event was seen in male patients after the first dose of mRNA vaccines, specifically BNT162b2 mRNA COVID-19 vaccine [1]. Indeed, in our study, 75.9% of ICSRs reported the hospitalization of the subjects with a mean duration of four days. This finding is not surprising if we consider that MIS-c represents a disease that may require hospitalization for the management of symptoms and complications associated with the disease. In some cases, MIS-c can be managed with symptom monitoring and symptomatic support at home, while in other cases hospitalization may be required for the administration of specific therapies, such as intravenous immunoglobulin and/or corticosteroids [25]. The duration of hospitalization for MIS-c varies according to the severity of the symptoms and complications associated with the disease. Literature data suggest that the median duration of hospitalization in MIS-c patients ranged from 6 to 12.5 days [26,27]. Kaushik et al. reported in their systematic review that all patients with MIS-c are hospitalized and the majority (N = 447; 68.0%) required a median duration of intensive care unit of five days (IQR, four to eight days) [28]. Furthermore, our analysis also provides insight into the time to onset of MIS-c following anti COVID-19 vaccination. Specifically, the median time to onset was 27 days (IQR, 5.0–59.1). MIS-c generally occurs between two to six weeks after the acute SARS-CoV-2 infection and data on the TTO of this event after anti COVID-19 vaccination are still scarce [3]. Literature data suggest that MIS-c can appear within 90 days after anti COVID-19 vaccination [29]. Furthermore, in line with our results, Yalçinkaya et al. reported the experience of a 12-year-old male who developed MIS-c 27 days after the first dose of the BNT162b2 mRNA COVID-19 vaccine [4]. Consistent with other real-world evidence from recently published pharmacovigilance studies, our results show a male predominance in MIS-c [30,31,32]. Despite data on sex differences in terms of the occurrence of MIS-c are still scarce, male predominance of MIS-c cases was discovered over time during the COVID-19 pandemic [33]. According to Chou et al., the mechanisms underlying genetic susceptibility to MIS-c depend on genetic predisposition, including polymorphisms or certain mutations [34]. Furthermore, it is known that male children are more likely to be affected by many infectious diseases than females and pathologies such as Kawasaki disease, which appears to be very similar to MIS-c in clinical presentation and symptomatology, are most frequently diagnosed in males [35,36,37]. Based on that, disproportionality analysis was performed in order to assess the probability of reporting ICSRs with MIS-c for males compared to females. Our findings suggested a higher reporting risk associated with males in each considered category. Despite this, our analysis is not designed for causal inference, it is not possible to conclude that a greater ROR in male individuals equals a higher rate of MIS-c. Consistent with available clinical evidence, our results show that pyrexia, which is usually defined as the first presenting sign of MIS-c, was the most commonly reported AEFI belonging to the SOC “General disorders and administration site conditions”. Moreover, C-reactive protein increase was the most frequently reported AEFI among the “Investigation” SOC, in line with the blood test predictors for the diagnosis of MIS-C. These events have been well-described together with MIS-c; specifically, C-reactive protein is an acute inflammatory protein that increases up to 1000-fold at sites of infection or inflammation and often precedes the onset of fever [38]. Considering the importance of clinical laboratory investigations in the differential diagnosis of patients presenting with similar conditions, we decided to analyze High Level Terms related to the most reported SOC “Investigations”.

Laboratory markers are indicative of a hyperinflammation process which appears to correlate with the severity of MIS-c. According to the findings in our study, one of the parameters to be evaluated in a patient with MIS-c is coagulation disorders and bleeding, which are often characterized by elevated values of D-dimers, prolonged Prothrombin Time (PT), and Partial Thromboplastin Time (PTT) [39]. Furthermore, levels of protein in the patient’s plasma affected by MIS-c are significantly dysregulated. Therefore, one of the first investigations that are carried out concerns the dosage of C reactive protein (CRP) that in cases of MIS-c has very high levels [40]. Generally, the normal CRP level in children is less than 0.6 milligrams per deciliter (mg/dL); in cases of MIS-c values were recorded around 3 until you reach 157 [41,42,43]. Finally, we analyzed the most reported PTs stratified by sex excluded from the SOC “Investigations”. As expected, although the symptoms manifested by children vaccinated against COVID-19 are characteristic of MIS-c, there are no sex differences with the most common symptoms manifested by the pediatric population following anti COVID-19 vaccination. Indeed, the most frequently reported AEFIs following anti COVID-19 vaccines were pyrexia, vomiting associated with abdominal pain, headache and diarrhea [44,45]. However, our results showed greater cardiac involvement for males than females with increased reporting of chest pain, left ventricular dysfunction, tachycardia, coronary artery dilatation and sinus tachycardia.

Lastly, it has to be considered that, as described above, it has been already demonstrated that MIS can be diagnosed in children previously contracted SARS-CoV-2 infection, regardless the immunization, and this explains why the Brighton Collaboration Group case definition, specific for MIS-c, requires the presence of “laboratory confirmed SARS-CoV-2 infection” or “personal history of confirmed or suspected COVID-19” or “close contact with known or suspected COVID-19 case” [3,19]. That confirmed relationship, between SARS-CoV-2 infection and the occurrence of MIS-c, therefore, makes it difficult, considering our data source, which is a pharmacovigilance database specific for the collection of adverse events following immunization, to distinguish the potential role of infection versus vaccination with respect to the onset of the study event. Moreover, pharmacovigilance databases are affected by limitations such as inaccuracy or incompleteness of information which enable us to take into consideration, in the analysis, the time to onset of MIS-c in order to attempt to distinguish between MIS-c induced by immunization and those induced by the infection alone. These considerations could explain why, to date, MIS-c has not been reported in the SmPC of mRNA COVID-19 vaccines, seeing the difficulty in verifying if the immunization could represent a risk factor for that syndrome. However, some clinical findings potentially suggest a possible correlation between immunization against SARS-CoV-2 and MIS-c occur [27,28,29].

Therefore, it is extremely important to continue monitoring the impact of vaccination and the risk of MIS-c.

## 4. Materials and Methods

### 4.1. Data Source

On 17 February 2023, data on Individual Case Safety Reports (ICSRs) related to the pediatric population (aged 0–20 years old) were retrieved from the Vaccine Adverse Event Reporting System (VAERS) database [46]. We searched VAERS reports from 1 January 2020 to 31 December 2022, for all vaccines, regardless of any characteristics such as race/ethnicity. The following information was considered for each ICSR: (a) patient characteristics (i.e., sex, age, co-medications, illness at the time of vaccination, pre-existing condition, and allergies to medications, food, or other products), (b) data about the suspected vaccines (i.e., vaccine name and type, manufacturer, manufacturer’s vaccine lot, number of doses administrated, route of administration, site, vaccination date, and prior vaccination event information), and (c) the AEFIs (i.e., description, onset date, outcome, and relevant diagnostic tests/laboratory data). The reported AEFIs are coded as Preferred Term (PT) according to the Medical Dictionary for Regulatory Activities (MedDRA) [47]. These data were originally included in three separate datasets: (i) “VAERSDATA”, which contains data on patients and about the outcome of the AEFIs, with their related description (narrative field), laboratory/diagnostic data, (ii) “VAERSVAX”, which provides the vaccine information, and (iii) “VAERSSYMPTOMS”, which provides the AEFIs coded according to the MedDRA dictionary. After downloading the three datasets from the VAERS website in comma-separated value (CSV) format, we created our own dataset by using the ICSRs’ identification codes as a unique key between the three above-mentioned datasets listed above. Moreover, through the analysis of the narrative fields, it was also possible to retrieve additional valuable information such as patient medical history, current illness, co-medications, the clinical course of the adverse events, seriousness and outcome of the events, laboratory results, action taken, and dechallenge and rechallenge information (when available).

### 4.2. Case Assessment

VAERS represents a passive national surveillance system. Spontaneous (or passive) surveillance means that clinicians, patients, and vaccine manufacturers send spontaneously their report of any AEFIs related to any U.S. FDA-approved vaccine. One of the main objectives of pharmacovigilance studies on spontaneous data is to identify early potential safety signals and generate hypotheses concerning possible new vaccine adverse events that were not identified in clinical trials. Specifically, we searched all ICSRs with coding or free text mention of potential cases MIS-c, which reported the following PT: “MIS-C”, “MISC”, “MIS”, “Multisystem Inflammatory Syndrome”, “Multisystem inflammatory”, “Multisystem inflammation”, “Multysistem + inflammation” and “Multysistem + inflammatory”. So, after selecting all ICSRs of suspected MIS-c following anti COVID-19 vaccination and removing any duplicates, we assessed them through the case definition and classification of the Brighton Collaboration in order to minimize the uncertainty of the selected cases [48]. This case definition was developed by a group of experts in the context of the active development of anti COVID-19 vaccines and other emerging pathogens. The definitions have been formulated with 3 levels of certainty: Level 1 (Definitive case), which is highly specific for the identification of a case of MIS-c, Level 2a or 2b (Probable case), Level 3a or 3b (Possible case). There are also two more levels: Level 4 (Insufficient Evidence) and Level 5 (Not a case of MIS-c).

The Level 1 classification requires (1) the presence of fever ≥ 3 consecutive days and, (2) 2 or more of clinical features such as rash, abdominal pain, hypotension, and headache, (3) laboratory marker of inflammation such as elevated ferritin or procalcitonin, (4) 2 or more measures of disease activity (e.g., neutrophilia or evidence of cardiac involvement by echocardiographic, and (5) laboratory-confirmed SARS-CoV-2 infection or personal history of confirmed COVID-19 within 12 weeks or close contact with known COVID-19 case within 12 weeks or following SARS-CoV-2 vaccination.

The Level 2a criteria are the same as the Level 1 criteria, with the exception of one measure of disease activity and within 12 weeks of a personal history of known or strongly suspected COVID-19 or close contact with a person with known or strongly suspected COVID-19 or following SARS-CoV-2 vaccination. Level 2b can be reached by the presence of fever for 1–2 consecutive days; otherwise, provide the same as Level 1.

Level 3a requires (1) the presence of fever ≥3 consecutive days and, (2) 2 or more clinical features, (3) no laboratory markers of inflammation or measures of disease activity available, (4) other measures of disease activity not available, and (5) within 12 weeks of a personal history of known or strongly suspected COVID-19 or close contact with a person with known or strongly suspected COVID-19 or following SARS-CoV-2 vaccination. Level 3b includes the same criteria as Level 2a except fever lasting 1–2 days and can be subjective.

Finally, Level 4 represents a reported event of MIS-c but with insufficient evidence to meet Levels 1, 2, or 3 of the case definition, and Level 5 is a non-case. Only the first three levels of certainty (“Definitive”, “Probable”, and “Possible” cases) have been included in the present study.

Despite using the Brighton Collaboration criteria, given the nature of the data source, we are unable to establish a causal link between the vaccine and MIS-c. In fact, we consider AEFI as “any adverse medical incident that occurs post-vaccination and is not necessarily linked to vaccination” [49].

### 4.3. Descriptive Analysis

A descriptive analysis was performed to evaluate sex-related differences in MIS-c following anti COVID-19 vaccination. Information on the total number of cases for individuals < 21 years old, the number of cases split for sex and the median age, the frequency and seriousness criteria of AEFIs, the frequency of the suspected/concomitant drugs other than anti COVID-19 vaccines were provided for all ICSRs. Moreover, we reported the data on the dose number administered, and the outcome (classified as “Fatal”, “Life-threatening”, “Disability”, and “Hospitalization”). Regarding the cases including the outcome of “Hospitalization”, it was also considered the hospital stay (number of days hospitalized), when available. The median time to event onset (TTO) was also assessed. The AEFIs were analyzed and then described by System Organ Classes (SOCs) and stratified by sex.

Finally, we estimated the rate of occurrence of COVID-19 vaccination-induced MIS-c with a 95% Confidence Interval (95% CI). To do this, we extracted data from CDC filtering by date (until 31 December 2022) and age (<21 years old).

All data management and statistical analysis were performed using R Statistical Software (R Foundation for Statistical Computing, Wien, Austria).

### 4.4. Disproportionality Analyses

A disproportionality analysis was performed to assess if males have a lower/higher probability of reporting ICSRs with MIS-c compared with females before and after the application of the algorithm developed by the Brighton Collaboration. Therefore, the Reporting Odds Ratio (ROR), with a 95% Confidence Interval (95% CI) was computed. The *p* value < 0.05 was used for statistical significance. All statistical analysis was performed using R Statistical Software (version 4.0.3; R Foundation for Statistical Computing, Wien, Austria).

## 5. Conclusions

Pre-marketing clinical trials for mRNA COVID-19 vaccines did not report cases of MIS-c following vaccination. Indeed, MIS has been reported as a late manifestation of SARS-CoV-2 infection in the pediatric population. To date, the potential severity of MIS-c and its possible long-term sequelae represent a valid reason to vaccinate children; indeed, literature data suggest that mRNA COVID-19 vaccines decreased the likelihood of MIS-c [18,50]. At the same time, recent studies report suspect cases of patients developing MIS-c after anti COVID-19 vaccine administration [3,4,5,6,7]. Our findings, in line with other recent studies, highlighted that mRNA COVID-19 vaccines could be potentially associated with MIS-c; such event seems to mostly involve young males, especially after the first dose. However, to date, Regulatory Agencies concluded that there is currently insufficient evidence on a possible link between anti COVID-19 vaccines and very rare cases of MIS [51]. Moreover, according to the literature, MIS-c related to COVID-19 vaccination may represent a milder phenotype compared with MIS-c secondary to SARS-CoV-2 infection [52]. Considering that the mechanisms underlying MIS-c following anti COVID-19 vaccination are poorly understood, it is clear that further investigations are needed also to better understand the long-term outcomes that will only emerge during the long-term follow-up. Furthermore, considering the involvement of different systems (cardiac, respiratory, gastrointestinal, and vascular), a multidisciplinary approach is essential to ensure a personalized treatment in relation to the severity of the multi-system involvement. Indeed, it is essential to increase the sensitivity above all of pediatricians to allow for early diagnosis and evaluate different treatment options for MIS-c in collaboration with other health professionals. Regarding our sex-stratified analysis, even if our findings suggested that males have a greater reporting risk, our analysis is not designed for causal inference, thus we cannot conclude that a higher reporting risk in males equals a higher rate of MIS-c.

This study has strengths and limitations. An important strength of this study was the use of the VAERS database which represents a useful and inexpensive tool for the collection and analysis of vaccine safety data. This database provides to better characterization of vaccine safety profiles and overcoming intrinsic limits of clinical trials. Indeed, a spontaneous reporting system allow the identification of specific AEFIs, not detectable during the pre-marketing phase, including rare and serious AEFIs, can be easily identified. Furthermore, the pharmacovigilance reporting system involves ICSRs related to the pediatric population, which are usually excluded by the pre-marketing clinical trials. However, the spontaneous reporting system is affected by limitations that are mainly related to the underreporting and poor quality or incompleteness of information listed in each ICSR. Therefore, we cannot exclude that important information was not listed in the ICSRs that we have evaluated (e.g., the time between infection and vaccination, comorbidities, the severity of the underlying illness, a detailed description of the adverse events, or a follow-up). Moreover, since the data were extracted from an American database, we were only able to include vaccines approved in the United States. Therefore, we lack data on vaccines approved in Europe and other parts of the world. Finally, as a pharmacovigilance analysis, it is important to acknowledge that our study does not establish a causal relationship between the vaccine and the adverse events.

Given that anti COVID-19 vaccines have just recently received commercial approval, further studies are urgently required to assess their safety profile and to a better understanding of sex-related differences in AEFIs. In this context, we believe that an effective pharmacovigilance system allows a continuous safety profile monitoring of these vaccines, providing to better define the risk/benefit profile of the anti COVID-19 vaccines and confirm or refute any possible association between mRNA COVID-19 vaccines and MIS-c. Furthermore, post-marketing studies can also help to identify sex-related differences in adverse events.

## Figures and Tables

**Figure 1 pharmaceuticals-16-01231-f001:**
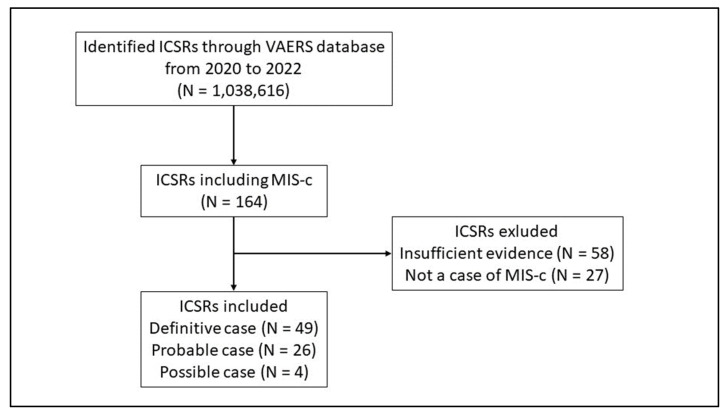
Flowchart for Individual Case Safety Reports inclusion and exclusion criteria. ICSRs have been classified as Level 4 in case of reported events with insufficient evidence to meet Level 1 (Definitive case), Level 2 (Probable case), and Level 3 (Possible case); Level 5 is a non-case of MIS-c.

**Figure 2 pharmaceuticals-16-01231-f002:**
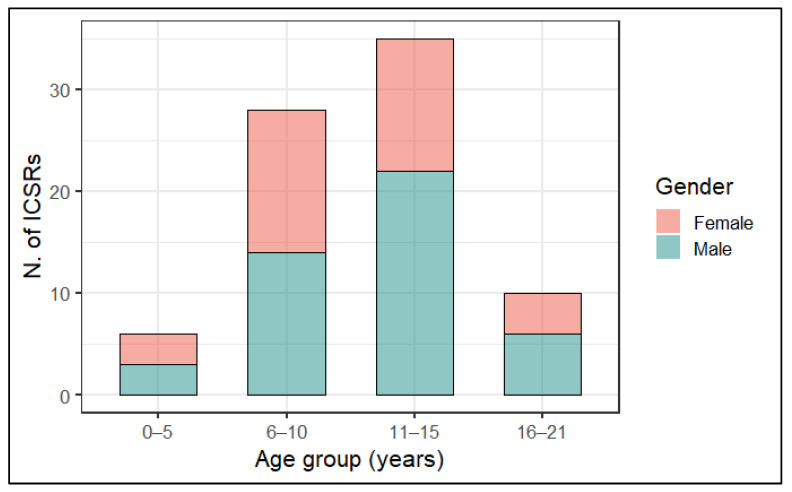
Distribution of cases of MIS-c following anti COVID-19 vaccination with stratification by age group and sex.

**Figure 3 pharmaceuticals-16-01231-f003:**
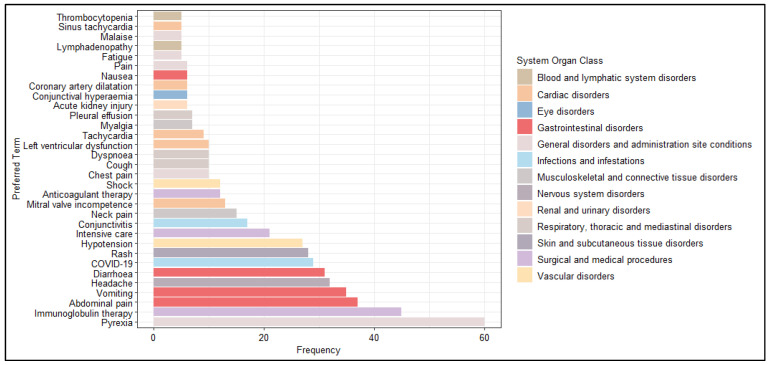
Distribution of Preferred Terms (PTs) (occurred at least 5 times) excluded from the System Organ Class (SOC) Investigations.

**Figure 4 pharmaceuticals-16-01231-f004:**
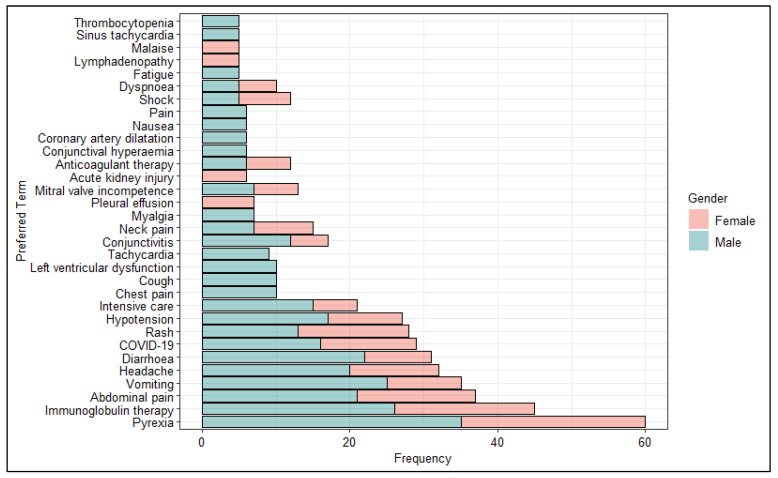
Distribution of Preferred Terms (PTs) (occurred at least 5 times) excluded from the System Organ Class (SOC) Investigations stratified by sex.

**Table 1 pharmaceuticals-16-01231-t001:** Demographic characteristics of Individual Case Safety Reports (ICSRs) reporting cases of MIS-c following anti COVID-19 vaccination reported in VAERS from 1 January 2020, to 31 December 2022 with stratification by sex.

	Female (%)	Male (%)	All ICSRs *
**N. ICSRs**	34 (43.0)	45 (57.0)	79 (100.0)
**Median age (IQR ^a^)**	10.5 (7.2–13.7)	10 (10.0–11.4)	11 (8.0–11.1)
**Outcome**			
Fatal	1 (2.9)	0 (-)	1 (1.3)
Life Threatening	8 (23.5)	10 (22.2)	18 (22.8)
Hospitalization	25 (73.6)	35 (77.8)	60 (75.9)
Median days (IQR)	4 (2–6)	4 (2–6)	4 (2–6)
**Vaccine**			
Comirnaty	34 (100.0)	43 (95.6)	77 (97.4)
Spikevax	0 (-)	1 (2.2)	1 (1.3)
NA	0 (-)	1 (2.2)	1 (1.3)
**Dose**			
Post 1st dose	26 (76.5)	29 (64.4)	55 (69.6)
Post 2nd dose	10 (23.5)	8 (17.7)	18 (22.8)
Post 3th dose	0 (-)	1 (2.2)	1 (1.3)
NA	0 (-)	5 (15.7)	5 (6.3)
**Median TTO ^b^ (IQR)**	25 (5–57.7)	27 (5.5–76.5)	27 (5.0–59.1)

* Individual Case Safety Reports. ^a^ IQR = interquartile range; ^b^ TTO = time to onset.

**Table 2 pharmaceuticals-16-01231-t002:** Distribution of Preferred Term (PT) (at least accounted for ≥3% of all AEFIs) belonging to the top 8 System Organ Classes (SOCs) with stratification by sex.

System Organ Class and Preferred Term	N. of Adverse Events (%)		
	Female (%)	Male (%)	Total (%)
*Investigations*	346 (100.0)	643 (100.0)	989 (100.0)
C-reactive protein increased	17 (34.0)	33 (66.0)	50 (5.1)
SARS-CoV-2 antibody test positive	17 (50.0)	17 (50.0)	34 (3.4)
Serum ferritin increased	12 (40.0)	18 (60.0)	30 (3.0)
*Gastrointestinal disorders*	50 (100.0)	95 (100.0)	145 (100.0)
Nausea and vomiting symptoms	15 (30.0)	29 (30.5)	44 (30.3)
Gastrointestinal and abdominal pains (excl oral and throat)	16 (32.0)	22 (23.2)	38 (26.2)
Diarrhoea (excl infective)	10 (20.0)	21 (21.1)	31 (21.4)
*General disorders and administration site conditions*	55 (100.0)	87 (100.0)	142 (100.0)
Pyrexia	26 (47.3)	38 (43.7)	64 (45.1)
Fatigue	5 (9.1)	9 (10.3)	14 (9.9)
Chest pain	3 (5.5)	9 (10.3)	12 (8.5)
Malaise	5 (9.1)	4 (4.6)	9 (6.3)
Pain	2 (3.6)	6 (6.9)	8 (5.6)
Chills	2 (3.6)	3 (3.4)	5 (3.5)
Illness	2 (3.6)	3 (3.4)	5 (3.5)
*Cardiac disorders*	26 (100.0)	76 (100.0)	102 (100.0)
Left ventricular dysfunction	2 (7.7)	10 (13.2)	12 (11.8)
Mitral valve incompetence	5 (19.2)	7 (9.2)	12 (11.8)
Tachycardia	2 (7.7)	8 (10.5)	10 (9.8)
Sinus tachycardia	3 (11.5)	6 (7.9)	9 (8.8)
Coronary artery dilatation	0 (-)	7 (9.2)	7 (6.9)
Pericardial effusion	3 (11.5)	4 (5.3)	7 (6.9)
Tricuspid valve incompetence	2 (7.7)	4 (5.3)	6 (5.9)
Cardiac dysfunction	1 (3.8)	3 (3.9)	4 (3.9)
Myocarditis	1 (3.8)	3 (3.9)	4 (3.9)
Right ventricular dysfunction	1 (3.8)	3 (3.9)	4 (3.9)
*Surgical and medical procedures*	36 (100.0)	55 (100.0)	91 (100.0)
Immunoglobulin therapy	19 (52.8)	29 (52.7)	48 (52.7)
Intensive care	7 (19.4)	14 (25.5)	21 (23.1)
Anticoagulant therapy	6 (16.7)	7 (12.7)	13 (14.3)
Endotracheal intubation	1 (2.8)	3 (5.5)	4 (4.4)
*Infections and infestations*	30 (100.0)	53 (100.0)	83 (100.0)
COVID-19	12 (37.5)	18 (35.3)	30 (36.1)
Conjunctivitis	5 (15.6)	17 (33.3)	22 (26.5)
Pneumonia	0 (-)	3 (5.9)	3 (3.6)
Upper respiratory tract infection	1 (3.1)	2 (3.9)	3 (3.6)
*Respiratory, thoracic and mediastinal disorders*	29 (100.0)	51 (100.0)	80 (100.0)
Cough	4 (14.8)	12 (22.6)	16 (20.0)
Dyspnoea	5 (18.5)	6 (11.3)	11 (13.8)
Pleural effusion	6 (22.2)	4 (7.5)	10 (12.5)
Oropharyngeal pain	3 (11.1)	4 (7.5)	7 (8.8)
Lung opacity	1 (3.7)	3 (5.7)	4 (5.0)
Pulmonary oedema	2 (7.4)	2 (3.8)	4 (5.0)
Atelectasis	1 (3.7)	2 (3.8)	3 (3.8)
Rhinorrhoea	0 (-)	3 (5.7)	3 (3.8)
*Immune system disorders*	32 (100.0)	39 (100.0)	71 (100.0)
Multisystem inflammatory syndrome in children	32 (100.0)	39 (100.0)	70 (98.6)

**Table 3 pharmaceuticals-16-01231-t003:** Disproportionality analysis.

Male vs. Female	ROR	95% CI	*p*-Value
MIS vs. no MIS (without BC)	3.07	2.24–4.20	<<0.005
MIS vs. no MIS (with BC)	2.69	1.72–4.20	<<0.005

## Data Availability

No new data were created to prepare this article, which was entirely based on the review of publicly available ICSRs retrieved from the EudraVigilance database.

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
