# Peer review of "Multisystem Inflammatory Syndrome in Children Following COVID-19 Vaccination: A Sex-Stratified Analysis of the VAERS Database Using Brighton Collaboration Criteria"

_pharmaceuticals, 2023, doi:10.3390/ph16091231_

Round 1

Reviewer 1 Report (Previous Reviewer 1)

The authors have addressed all the comments raised in previous revision. After addressing the comments, the current version of the manuscript quite good and suitable for publication.

Author Response

Dear Reviewer,

We would like to express our sincere gratitude for your acceptance of our manuscript. Your positive response is truly appreciated.

For your reference, the revised manuscript, incorporating the suggestions of the other reviewer, is attached. All modifications have been highlighted for your convenience.

Thank you once again for your valuable support.

Reviewer 2 Report (Previous Reviewer 2)

The title is changed to be more relevant to the material and methods and results.

Still, MIS-C connection with COVID-19 vaccines was not discussed as a conseqence, neither a causality was discussed.

The authors presented data on the adverse events following COVID-19 vaccination, and not on MIS-C, anf this should be emphasized in the text.

The conclusion of more men being affected by MIS-C based on these outcomes is not a valid argument. The article has severe flaws and speculations.

In the abstract - "As time goes by, MIS-c was also reported as a potential adverse event following anti COVID-19 vaccination." - please, cite the relevant sources for this statement

in the abstract" -  The present pharmacovigilance study aimed to describe MIS-c, stratified by sex, reported in the Vaccine Adverse Events Reporting System 24 (VAERS). - MIS-C should be diagnosed according to the criteria, not just assumed by the VAERS.

Line 52-53 - Please add: These cases suggest that a possible 51 association between the COVID-19 vaccine and MIS-c cannot be excluded, "but is not yet confirmed."

The paper has not been improved significantly. Still, the main hypothesis for MIS-C as a consequence of covid-19 vaccination is not discussed with cautious. Also, the design of the study is not appropriate.

Author Response

Dear Reviewer,

Thank you for your feedback, which has been instrumental in improving our manuscript. We have made all the suggested revisions, and these changes are highlighted in the uploaded file for your convenience.

Your thorough review has been immensely valuable, and we appreciate your dedication to enhancing the quality of our manuscript.

Here are our point-by-point responses addressing your specific comments.

Point 1: The title is changed to be more relevant to the material and methods and results.

Response 1: Thank you for your suggestions about the title.

Point 2: Still, MIS-C connection with COVID-19 vaccines was not discussed as a conseqence, neither a causality was discussed.

Response 2: In the introduction, we have added a passage that attempts to explain the rationale behind the reports of MIS-c following vaccination (lines: 58-63). However, it's important to note that the precise mechanism underlying MIS-c, whether arising from COVID-19 vaccination or COVID-19 infection itself, is not yet fully understood. Despite extensive research, the complexities of the immunological response involved in MIS-c are still being elucidated. This inherent uncertainty regarding the exact mechanisms remains a topic of ongoing investigation in the field.

Point 3: The authors presented data on the adverse events following COVID-19 vaccination, and not on MIS-C, anf this should be emphasized in the text.

Response 3: We updated the abstract to be more specific about our findings, referring to "MIS-c following COVID-19 vaccination" rather than "MIS-c". Moreover, new periods have been added (lines: 58-63) to emphasize our interest on MIS-c following COVID-19 vaccination.

Point 4: The conclusion of more men being affected by MIS-C based on these outcomes is not a valid argument. The article has severe flaws and speculations.

Response 4: Our hypothesis underpinning our analysis was that there could be a sex difference in the onset of MIS-c following COVID-19 vaccination, akin to the observed variance in incidence and severity of MIS-C post SARS-CoV-2 infection. Anyway, our analysis is not designed for causal inference, so it is not possible to conclude that a greater ROR in male individuals equals a higher rate of MIS-c. To emphasize this point, a new period has been added (lines 252-253).

Point 5: In the abstract - "As time goes by, MIS-c was also reported as a potential adverse event following anti COVID-19 vaccination." - please, cite the relevant sources for this statement

Response 5: In the abstract, the statement "As time goes by, MIS-C was also reported as a potential adverse event following anti COVID-19 vaccination" is supported by references to reports from authoritative sources such as the Centers for Disease Control and Prevention (CDC), the Pharmacovigilance Risk Assessment Committee (PRAC), and the collection of case reports documented in the manuscript itself.

Point 6: in the abstract" -  The present pharmacovigilance study aimed to describe MIS-c, stratified by sex, reported in the Vaccine Adverse Events Reporting System 24 (VAERS). - MIS-C should be diagnosed according to the criteria, not just assumed by the VAERS.

Response 6: In the abstract, we have included a clarification that the cases described in the study indeed meet the diagnostic criteria outlined by the Brighton Collaboration for Multisystem Inflammatory Syndrome in Children (MIS-C) (lines: 25-26).

Point 7: Line 52-53 - Please add: These cases suggest that a possible 51 association between the COVID-19 vaccine and MIS-c cannot be excluded, "but is not yet confirmed."

Response 7: The statement you mentioned has been incorporated into the manuscript (now line 55).

Point 8. The paper has not been improved significantly. Still, the main hypothesis for MIS-C as a consequence of covid-19 vaccination is not discussed with cautious. Also, the design of the study is not appropriate.

Response 8: We appreciate your continued feedback and thorough review of our manuscript. Your insights have been incredibly valuable in refining our work. We believe that as a result of your suggestions, the manuscript has indeed undergone significant improvements. Regarding your concern about the main hypothesis of MIS-c as a consequence of COVID-19 vaccination, we have taken a more cautious approach in discussing it. We have added additional information in the Introduction (lines: 58-63, 100-101), Discussion (204-211, 252-253), and Methods (392-395) to better elucidate certain aspects and limitations of the analysis. Furthermore, we would like to emphasize that the primary objective of our study was to identify cases of MIS-c following COVID-19 vaccination, describe their characteristics, and explore potential sex differences. The design of the study was tailored to achieve these objectives. We acknowledge that every study design has inherent limitations, and we have strived to transparently address these in the revised manuscript.

Round 2

Reviewer 2 Report (Previous Reviewer 2)

The paper has been improved significantly in their design and presentation.

This manuscript is a resubmission of an earlier submission. The following is a list of the peer review reports and author responses from that submission.

Round 1

Reviewer 1 Report

1.      How many ICSRs were obtained from VAERS during the study period, and how many of them reported MIS-c as an adverse event in subjects aged < 21 years?

2.      What was the application of the Brighton Collaboration's algorithm used for, and how many ICSRs were excluded based on this algorithm?

3.      What were the demographic characteristics of the patients who experienced MIS-c, including gender distribution and median age?

4.      Which COVID-19 vaccine was most commonly reported as suspected in cases of MIS-c, and at what time point after vaccination did MIS-c mainly occur?

5.      What were the most common System Organ Classes (SOCs) and Preferred Terms (PTs) associated with MIS-c in the included ICSRs, and were there any gender differences in the distribution of these AEFIs?

Reviewer 2 Report

The authors failed to design a proper study. The title is not relevant to the material and methods and results. MIS-C connection with COVID-19 vaccines was not discussed as a conseqence, neither a causality was discussed. The authors presented data on the adverse events following COVID-19 vaccination, and not on MIS-C. The conclusion of more men being affected by MIS-C based on these outcomes is not a valid argument. The article has severe flaws and speculations.

Minor grammar and style editing is required.